# Does Reasoning Emerge? Examining the Probabilities of Causation in Large Language Models

**Javier González**
Gonzalez.Javier@microsoft.com
Microsoft Research, Cambridge

**Aditya V. Nori**
Aditya.Nori@microsoft.com
Microsoft Research, Cambridge

## Abstract

Recent advances in AI have been significantly driven by the capabilities of large language models (LLMs) to solve complex problems in ways that resemble human thinking. However, there is an ongoing debate about the extent to which LLMs are capable of actual *reasoning*. Central to this debate are two key probabilistic concepts that are essential for connecting causes to their effects: the probability of necessity (PN) and the probability of sufficiency (PS). This paper introduces a framework that is both theoretical and practical, aimed at assessing how effectively LLMs are able to replicate real-world reasoning mechanisms using these probabilistic measures. By viewing LLMs as abstract machines that process information through a natural language interface, we examine the conditions under which it is possible to compute suitable approximations of PN and PS. Our research marks an important step towards gaining a deeper understanding of when LLMs are capable of reasoning, as illustrated by a series of math examples.

## 1 Introduction

Large language models (LLMs) have revolutionized the way we interact with technology, enabling more natural and intuitive communication between humans and computers in applications like writing assistants [8], sentiment analysis in social media [29], healthcare [10, 35] and many others. Despite the surge of interest and recent breakthroughs [5], the ability of LLMs to *reason* about real-world problems as humans do continues to be a topic of intense research [1, 14].

Reasoning is a cognitive process that involves drawing conclusions, making judgments, or forming inferences based on facts or premises. This process has been explored from various perspectives. *Symbolic reasoning* [17] involves the manipulation of symbols that represent ideas or objects and it is often used in mathematics and logic to represent numerical values or logical propositions. *Causal reasoning* [26] focuses on discerning the relationship between a cause and its effect, aiming to understand how certain events can impact other. Other forms of reasoning include *inductive reasoning* [7] (making broad generalisations from specific observations), *deductive reasoning* [27] (applying general principles to specific cases), and *abductive reasoning* [2] (forming the best hypothesis based on incomplete information).

In the realm of LLMs, reasoning is typically understood to be the ability of these models to demonstrate *emergent* capabilities that surpass mere statistical pattern recognition in the training set. It entails systematically breaking down problems into a logical sequence of smaller, manageable steps and then processing these steps internally to arrive at accurate conclusions that are grounded in reality. This concept is the foundation for techniques such as *chain of thoughts prompting* [34], which aim to

38th Conference on Neural Information Processing Systems (NeurIPS 2024).

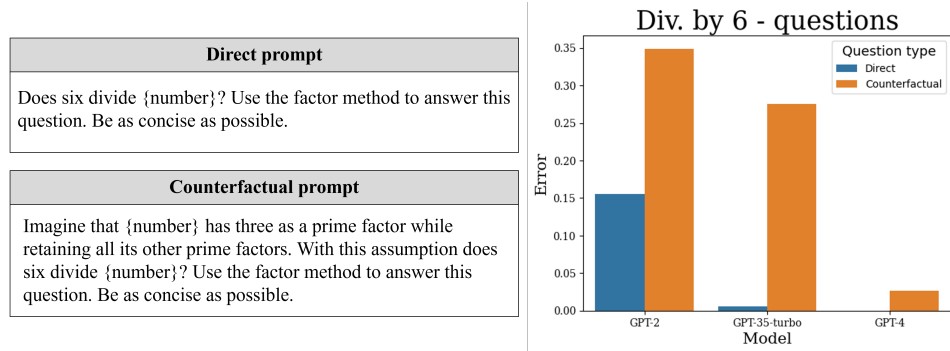

Figure 1: Illustration of the actual vs. perceived reasoning abilities of GPT-2, GPT-35-turbo and GPT-4 for a simple arithmetic problem. We posed two distinct types of questions (direct and counterfactual) to the models, each repeated 10 times, for every {number} from 1 to 50. All three models showed an inflated sense of reasoning capability when answering the direct questions. The discrepancy is especially pronounced in GPT-35-turbo, which performed nearly flawlessly on direct questions, but experienced a surge in error rate, exceeding 25%, when handling counterfactual questions.

teach LLMs how to reason by providing examples where problems are solved through a sequence of smaller steps.

Assessing the reasoning abilities of LLMs involves distinguishing between two aspects: the accuracy with which an LLM solves a problem, and its capacity to understand and process the fundamental elements that lead to that solution. Judea Pearl, in his hierarchy of causality [25], asserts: *"Only machines that can correctly perform correlations, interventions and counterfactuals will have reasoning abilities comparable to human"*. As demonstrated in [16], while LLMs are remarkable in using learnt patterns from their training data to generate correct answers (correlations), they falter when faced with hypothetical/imaginary scenarios that were not part of their training (counterfactuals). This is depicted in Figure 5, which presents a straightforward arithmetic problem (this is the **direct prompt** in the figure). Both GPT-35-turbo and GPT-4 can accurately determine the divisibility of numbers by 6, suggesting at first glance that they can reason about divisibility. However, when the questions are framed in a counterfactual manner (this is the **counterfactual prompt** in the figure), only GPT-4 maintains a low error rate, indicating its superior ability to handle such reasoning tasks.

In this paper, we introduce a systematic method to assess the reasoning capabilities of LLMs by examining the concepts *necessity* and *sufficiency*, which are key elements of logical reasoning and have been studied in multiple fields such as logic, probability, and causality [22, 19, 12]. In propositional logic, a sufficient condition is defined as $X \implies Y$, indicating that the presence of $X$ ensures the occurrence of $Y$. On the other hand, a necessary condition is defined as $Y \implies X$, signifying that the occurrence of $Y$ necessitates the prior occurrence of $X$. We focus on the probabilistic interpretations of necessity and sufficiency [24]. The *probability of necessity* (PN) between two boolean variables $X$ and $Y$ is defined as $\mathrm{PN}(x, y) := \mathbb{P}(y'_{x'}|x, y)$. Here, $y'_{x'}$ represents the counterfactual value of $Y = y'$, had $X$ been set to a different value $x'$. By conditioning on both $X = x$ and $Y = y$, this measure captures probability of observing a different outcome in the absence of the event $X = x$. The *probability of sufficiency* (PS), on the other hand, is defined as $\mathrm{PS}(x, y) := \mathbb{P}(y_x|x', y')$ and measures the probability that $X = x$ results in $Y = y$, for cases where both originally had different values.

We show that when a problem can be solved via a reasoning graph of boolean conditions, denoted by $\mathcal{G}$, the PN and PS can be computed using a causal model underlying $\mathcal{G}$. As described in [24], the exact computation of PN and PS requires samples from the (causal) data generative model, counterfactual data (experiments) as well as other monotonicity assumptions. As a *reasoning test*, we statistically compare the true PN and PS measures (computed by sampling from the original and the intervened graph) with those simulated via factual and counterfactual datasets generated by an LLM. Figure 2 presents an informal illustration of the reasoning test advocated in this paper, focusing on the specific problem of determining whether a number $N$ is divisible by 6. The test relies on the reasoning principle that: *"A natural number $N$ that is divisible by both 2 and 3 is also divisible by 6"*. This logic is represented in a reasoning graph $\mathcal{G}$ that links the conditions $C_2$ (divisibility by 2) and $C_3$

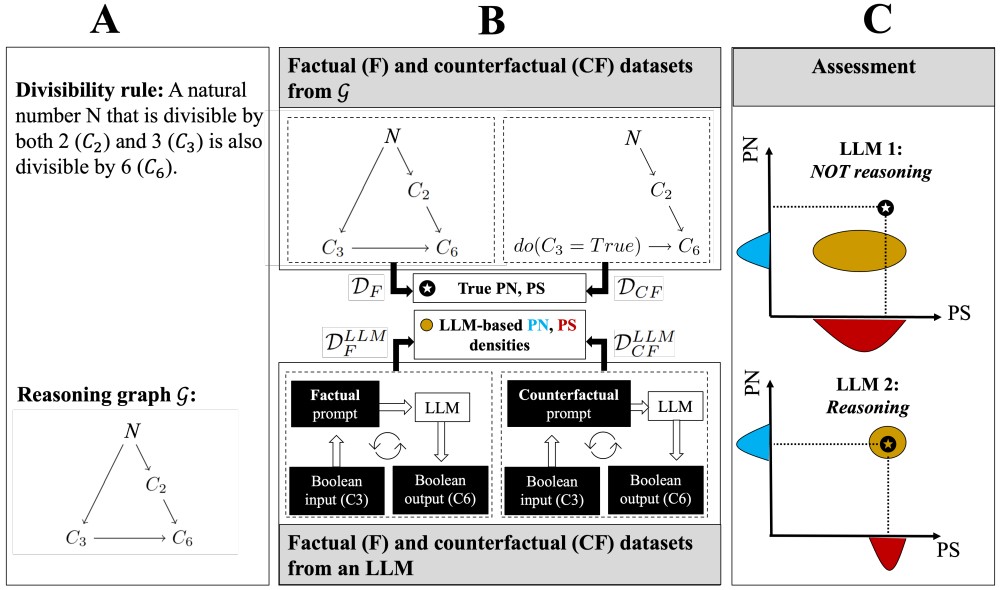

Figure 2: Reasoning test for assessing an LLM's reasoning abilities. **A)** Divisibility rule and the corresponding reasoning graph. **B)** Dataset generation for computing PN and PS. **C)** Analysis comparing actual values of PN and PS with PN and PS estimates for the LLM-generated data.

(divisibility by 3) to the conclusion $C_6$ (divisibility by 6). We test the reasoning abilities of an LLM using natural numbers $N$ from 1 to 400. This is shown in Figure 2(A).

As indicated in Figure 2(B), we create two sets of data based on $\mathcal{G}$. The first is a factual dataset ($\mathcal{D}_F$) which captures whether each number $N$ satisfies conditions $C_2$ and $C_3$. The second is a counterfactual dataset, ($\mathcal{D}_{CF}$), which assumes condition $C_3$ is always true and then records whether each number $X$ would satisfy $C_6$ under this assumption/intervention (realised by $do(C_3 = True)$ in the figure). For the LLM being evaluated, we also produce two datasets. The first, $\mathcal{D}_F^{LLM}$, documents the LLM's response for $C_6$ for each number $N$, when the prompt is based on the reasoning graph $\mathcal{G}$. The second, $\mathcal{D}_{CF}^{LLM}$, involves a hypothetical scenario where we assume $C_3$ is true and then record the LLM's prediction for $C_6$ given this "counterfactual prompt". This process is repeated multiple times (several answers from the LLMs are collected from each prompt). We assess the LLM's reasoning capability by comparing the estimated (distribution of) PN and PS from the $\mathcal{D}_F^{LLM}$ and $\mathcal{D}_{CF}^{LLM}$ datasets with the actual values derived from $\mathcal{D}_F$ and $\mathcal{D}_{CF}$ datasets. Figure 2(C) displays these comparisons, plotting PN vs. PS. The closer the estimated PN/PS values to the actual PN/PS values, the better it is at reasoning. In this case, LLM 2 demonstrates better reasoning abilities than LLM 1.

**Related work:** Reasoning in LLMs has been studied from multiple perspectives. [15] presents an overview paper that elucidates key reasoning concepts utilised by LLMs. [13] examines the similarity between reasoning with a language model and planning with a world model, proposing a novel reasoning framework that redefines the LLM as both a world model and a reasoning agent. Various studies [28, 4] have focused on assessing the reasoning and problem-solving abilities of LLMs, yet none have used the probabilities of causation as the primary objects of computation as done in our research. [32] carries out a series of experiments to show that LLMs can indeed derive benefits from reasoning errors, offering potentially cost-effective strategies by using mistakes to bolster reasoning capabilities. Recent research indicates that LLMs like GPT-3.5 and GPT-4 are effective at causal reasoning tasks, including pairwise causal discovery [18]. These models have achieved state-of-the-art performance on multiple causal benchmarks, outperforming existing algorithms. Nevertheless, LLMs also exhibit unpredictable failure modes, and currently, they are not capable of discovering new knowledge or making high-stakes decisions with a high level of precision [20, 36, 21, 6, 3].

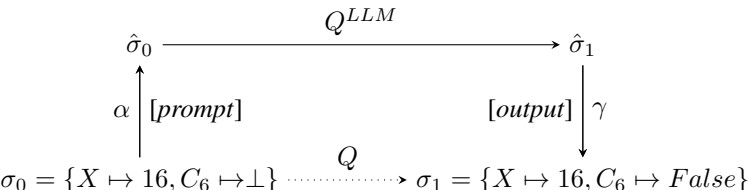

Figure 3: The HEX diagram depicts two approaches for solving the problem $(Q, \sigma_0)$ outlined in Example 1. The dotted path corresponds to the actual process of solving the problem, while the solid path represents the one taken by the LLM.

**Contributions:** This paper presents two main contributions.

i) A novel, theoretical and practical, framework to evaluate the reasoning capabilities of language models using the probabilistic of causation, specifically the probability of necessity and the probability of sufficiency. Our approach is unique in the sense that it allows to differentiate generalization by reasoning from merely replicating statistical patterns in the training data.

ii) Empirical tests on various reasoning problems as well as several insights about the reasoning abilities of language models in the GPT family.

## 2 LLMs as abstract machines

As described by the HEX framework [9], an LLM functions as an abstract machine that uses natural language as an interface. In this section, we introduce the core elements of this framework, which will subsequently enable us to define an LLM's internal representation of PN and PS.

We define a *problem* as a query-state pair $(Q, \sigma)$. The state $\sigma$ is a mapping defined by $\sigma : \mathcal{V} \to \mathcal{C}$, which assigns values from a specified domain $\mathcal{C}$ to a set of variables $\mathcal{V} = \{V_1, \ldots, V_n\}$. The query $Q : 2^{\mathcal{V} \to \mathcal{C}} \to 2^{\mathcal{V} \to \mathcal{C}}$ is a mapping that transforms an input state $\sigma$ to a well-defined output state. To *solve a problem* is to calculate $\sigma_1 = Q(\sigma_0)$, where $\sigma_0$ and $\sigma_1$ represent the states before and after the query $Q$ is applied. To clarify this, we consider the following example:

**Example 1.** *"Given that a natural number divisible by both 2 and 3 is also divisible by 6, determine whether the number 10 is divisible by 6."*

To solve Example 1, we apply the query $Q$ to the state $\sigma_0 = \{N \mapsto 10, C_6 \mapsto \perp\}$, where $Q = \lambda\sigma . (\sigma(N) \pmod 2) \equiv 0) \wedge (\sigma(N) \pmod 3) \equiv 0)$[1]. This results in a final state $\sigma_1 = \{N \mapsto 10, C_6 \mapsto False\}$, thereby resolving the problem with $\sigma_1(C_6) = Q(\sigma_0) = False$.

We now turn to the question of how an LLM solves a problem defined by a query-state pair $(Q, \sigma_0)$. This process involves three essential steps as illustrated by Figure 3:

1. First, an abstraction mapping translates the initial state $\sigma_0$ into a latent state $\hat{\sigma}_0$ via a *prompt*.
2. Next, the LLM processes (via the query $Q^{LLM}$) this latent state $\hat{\sigma}_0$.
3. Finally, the output mapping transforms the LLM output latent state $\hat{\sigma}_1$ back into a concrete state, producing the final *output* $\sigma_1$.

Formally, solving a problem $(Q, \sigma_0)$ with an LLM can be described as a sequence of function applications resulting in the output $\sigma_1 = (\gamma \circ Q^{LLM} \circ \alpha)(\sigma_0)$. To illustrate this, the problem statement is given as a prompt input to GPT-4 [23]. The response from GPT-4 is "False", which matches the result obtained by applying the query $Q$ directly to the input state $\sigma_0$. When both the direct application of $Q$ and the LLM computation yield the same answer, we say that the diagram, as shown in Figure 3, is commutative–meaning that following either the dotted line or the solid lines lead to the same result. For a more in-depth explanation of this framework, please refer to [9].

---

[1]See https://en.wikipedia.org/wiki/Lambda_calculus for a quick introduction to Lambda calculus.

# 3 Probabilities of causation for an LLM

To assess the reasoning abilities of an LLM, we must link its generated responses to the actual reasoning processes that produced those responses. For a problem $(Q, \sigma)$, we postulate the existence of a causal model $\mathcal{M}_\mathcal{V}$ defined over variables in $\mathcal{V}$, and by a set of structural equations and endogenous variables. For a detailed introduction to causal models, refer to Appendix A. Additionally, the seminal work by Pearl [25] on causality provides foundational insights on this area. Here, we are particularly interested in causal models that represent the logical steps involved in problem-solving. However, it is important to note that the concept of a causal model is broadly applicable beyond this specific application.

We assume that $\mathcal{V} = \{X, Y, Z\}$, which includes $X$ and $Y$ as boolean variables, and $Z$ as a variable (which may be multivariate) that encompasses all necessary factors that are required to understand how an intervention on $X$ would affect $Y$. In the context of causality, this means that the distribution $\mathbb{P}(Y|do(X = x'))$, where $do$ denotes the intervention operator defined in [25], is identifiable. This means we can predict the outcome for $Y$, and that the counterfactual $Y_{X=x'}$, that can be read as *"the value of $Y$ had $X$ been $x'$"*, is well-defined. For further details, please refer to Appendix A. For ease of exposition in the following text, we will simplify our notation by omitting the explicit reference to $Z$. Therefore, we will denote $Y_{X=x}(Z = z)$ more succinctly as $Y_{X=x}$.

As studied in [31], if $Y$ is monotonic with respect to $X$, then PN and PS can be computed as follows:

$$\text{PN}(x, y) = \frac{\mathbb{P}(y) - \mathbb{P}(y|do(x'))}{\mathbb{P}(x, y)} \text{ and } \text{PS}(x, y) = \frac{\mathbb{P}(y|do(x)) - \mathbb{P}(y)}{\mathbb{P}(x', y')}. \tag{1}$$

To estimate PN and PS, we need two different types of datasets. The first is a *factual* dataset $\mathcal{D}_F = \{x_i, y_i, z_i\}_{i=1}^n$, which is used to infer $\mathbb{P}(y)$, $\mathbb{P}(x, y)$ and $\mathbb{P}(x', y')$. The second dataset $\mathcal{D}_{CF} = \{x_i, Y_{X=x_i}, z_i\}_{i=1}^n$ is a *counterfactual* one, and is necessary to determine $\mathbb{P}(y|do(x))$ and $\mathbb{P}(y|do(x'))$.

There are various methods to acquire the datasets $\mathcal{D}_F$ (factual) and $\mathcal{D}_{CF}$ (counterfactual). For a physical process, the usual method would be through observation and experimentation. However, in this paper, we presume access to a comprehensive reasoning graph that is equivalent to a causal model $\mathcal{M}_\mathcal{V}$. This allows us to simulate and generate the $\mathcal{D}_F$ and $\mathcal{D}_{CF}$ datasets. Both $\mathcal{M}_\mathcal{V}$ and the sub-model $\mathcal{M}_{\mathcal{V},do(X=x)}$ define two distinct joint probability distributions $\mathbb{P}_{\mathcal{M}_\mathcal{V}}$ and $\mathbb{P}_{\mathcal{M}_{\mathcal{V},do(X=x)}}$ over $X$, $Y$ and $Z$. We obtain the datasets $\mathcal{D}_F$ and $\mathcal{D}_{CF}$ by sampling from these respective probability distributions. These datasets are then used to calculate PS and PN using Eq. (1).

## 3.1 LLM-based counterfactuals

*Can an LLM reason in a manner that is consistent with $\mathbb{P}_{\mathcal{M}_\mathcal{V}}$?* In Example 1, we obtained consistent answers (that is, the corresponding HEX diagram commutes) for a direct divisibility question. However, to evaluate the reasoning abilities of the LLM, it is crucial that this consistency is also observed when the queries are framed in a counterfactual manner. This is necessary to ensure that the LLM can apply its reasoning to imaginary situations that are unlikely to be present in the training set, demonstrating its ability to generalise based on a correct internal representation of the reasoning logic of the problem. Practically, this means employing the LLM as a "counterfactual data simulator", where the data it generates under these hypothetical conditions are used to estimate PN and PS.

**Definition 1** (Counterfactual query). *Consider a problem $(Q, \sigma_0)$, with $\sigma_0 = \{X \mapsto x, Y \mapsto y, Z \mapsto z\}$ being an initial state. Let $\mathcal{M}_\mathcal{V}$ be a causal model over the variables $\mathcal{V}$. We can then define a counterfactual query $Q'$ as follows: $Q'(\sigma_0) = \{X \mapsto x', Y \mapsto Y_{X=x'}, Z \mapsto z\}$.*

In other words, a counterfactual query updates two variables of the state: it sets $X$ to its new value $x'$, and $Y$ to the counterfactual $Y_{X=x'}$. An LLM-based counterfactual $Y_{X=x'}^{LLM}$ is computed as follows:

$$Y_{X=x'}^{LLM} = (\gamma \circ Q'^{LLM} \circ \alpha)(\sigma_0)(Y)$$

where $\sigma_0 = \{X \mapsto x, Y \mapsto y, Z \mapsto z\}$, and $Q'^{LLM}$ is a counterfactual query. This entire process simulates counterfactual reasoning within the LLM, and is facilitated through textual prompts that are structured to elicit the desired counterfactual outcome.

**Definition 2** (Counterfactual prompt). *A counterfactual prompt is a textual encoding of a counterfactual query for some initial state $\sigma_0$.*

| $\mathcal{D}_F$ | $C_6=0$ | $C_6=1$ |
|---|---|---|
| $C_3=0$ | 267 | 0 |
| $C_3=1$ | 67 | 66 |
| $\mathcal{D}_{CF}$ | $C_6=0$ | $C_6=1$ |
| $C_3=0$ | 133 | 0 |
| $C_3=1$ | 133 | 134 |
| $\mathcal{D}_F^{GPT-4}$ | $C_6=0$ | $C_6=1$ |
| $C_3=0$ | 267 | 0 |
| $C_3=1$ | 67 | 66 |
| $\mathcal{D}_{CF}^{GPT-4}$ | $C_6=0$ | $C_6=1$ |
| $C_3=0$ | 132 | 1 |
| $C_3=1$ | 132 | 135 |

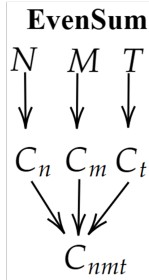

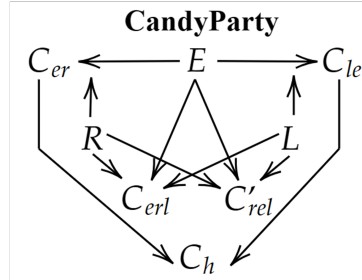

Figure 4: *Left*: Contingency tables for $\mathcal{D}_F$, $\mathcal{D}_{CF}$ and $\mathcal{D}_{CF}^{GPT-4}$ in Example 1. *Right*: Reasoning graphs for the other math problems in this paper. C-type nodes in the graph represent boolean conditions. See Appendix C for details.

Figure 1 shows an example of a counterfactual prompt. To create a comprehensive dataset $\mathcal{D}_{CF}^{LLM}$ of counterfactuals based on an LLM, we start with the factual dataset $\mathcal{D}_F^{LLM}$. From this dataset, we generate a set of initial states $\sigma_{0,i} = \{X \mapsto x_i, Y \mapsto y_i, Z \mapsto z_i\}$, which serve as the basis for deriving counterfactuals using the LLM. To compute PN and PS, we substitute $\mathcal{D}_F$ with $\mathcal{D}_F^{LLM}$ and $\mathcal{D}_{CF}$ with $\mathcal{D}_{CF}^{LLM}$ in Eq. (1).

**Example 1 revisited**. We construct four distinct datasets using every integer in $[1, 400]$: the factual dataset $\mathcal{D}_F$, the counterfactual dataset $\mathcal{D}_{CF}$, the LLM-based factual dataset $\mathcal{D}_F^{LLM}$, and the LLM-based counterfactual dataset $\mathcal{D}_{CF}^{LLM}$. These datasets, shown in Figure 4 (*Left*) are generated following the causal model shown in Figure 2, its modified version with interventions, and the LLM prompting methods mentioned previously. We obtain PN $= 1$ and PS $= 0.50$ for the datasets $\mathcal{D}_F$ and $\mathcal{D}_{CF}$. On the other hand, PN$^{GPT-4} = 0.984$ and PS$^{GPT-4} = 0.505$, when we use the factual $\mathcal{D}_F^{LLM}$ and counterfactual $\mathcal{D}_{CF}^{LLM}$ datasets generated by GPT-4.

### 3.2 Counterfactual consistency in LLMs

**Definition 3** ($\beta$-counterfactual consistency). *Consider a structural causal model $\mathcal{M}_\mathcal{V}$ with variables $\mathcal{V} = \{X, Y, Z\}$. Let $\mathcal{A}_{X=x}(Z)$ be a function that generates counterfactuals for $Y$. We say that $\mathcal{A}$ is $\beta$-counterfactual consistent with $\mathcal{M}_\mathcal{V}$ if the following condition is satisfied: $\mathbb{E}_{\mathbb{P}(X,Y,Z)}[\mathcal{A}_{X=x}(Z=z) \neq Y_{X=x}(Z=z)] \leq \beta$, where $\beta \leq 0$.*

$\beta$-counterfactual consistency defines the limit error rate for counterfactuals produced by $\mathcal{A}_{X=x}(Z = z)$. This error rate should ideally be zero for an LLM that exhibits flawless reasoning abilities. The following lemma specifies the conditions necessary for this property to hold (the proof can be found in Appendix D).

**Lemma 1.** *Let $\mathcal{M}_\mathcal{V}$, with variables $\mathcal{V} = \{X, Y, Z\}$, be a structural causal model for a problem $(Q, \sigma_0)$, and let $M$ be an LLM that generates counterfactuals for $Y$. Then $M$ is $\beta$-counterfactual consistent with $\mathcal{M}_\mathcal{V}$ if and only if its associated HEX diagram for the problem $(Q', \sigma_0)$, where $Q'$ is the counterfactual version of $Q$, is commutative for all admissible values of $X$, $Y$ and $Z$.*

Lemma 1, provides a theoretical criterion to test the consistency of an LLM with some reasoning graph described by $\mathcal{M}_\mathcal{V}$. As we show next in the experimental section, the commutability of the diagram is tested in expectation by sampling over the variables in $\mathcal{V}$.

## 4 Empirical illustrations

We focus on three math problems, each with a progressively higher level of difficulty.

*Divisibility by 6* (`Div6`): We compute the PN and PS to determine the impact that an integer $N$'s divisibility by 3 (denoted as $C_3$) has on its divisibility by 6 (denoted as $C_6$). For our analysis, we consider $N \in [1, 400]$.

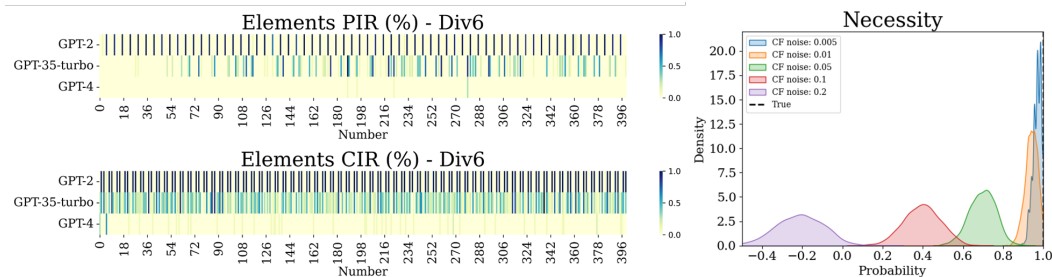

Figure 5: *Left*: Heatmaps comparing the consistency of data generated by GPT-2, GPT-3.5-turbo, and GPT-4 for the `Div6` problem. Each heatmap cell represents the error rate of the corresponding model for each element of the problem across 10 replicated tests. *Right*: Sensitivity of the simulated PN relative to varying levels of random noise introduced in the true counterfactuals.

*Even sum of integers* (`EvenSum`): We examine scenarios where the sum of three integers $M$, $N$ and $T$ is even. This can occur under two conditions: when all three integers are even, or when one is even and the other two are odd. We evaluate PN and PS for impact that $M$ being odd or even ($C_m$) has on the resulting sum being odd or even ($C_{mnt}$). For our analysis, we consider all possible values for $M$, $N$ and $T$, with each integer ranging from 1 and 8.

*Candy party* (`CandyParty`): In this hypothetical scenario, Rafa is having his birthday party with two guests, Lara and Emma. They have 20 candies to distribute among themselves. The party will be considered 'happy' if the candy distribution satisfies at least one of the following conditions: (i) Each person gets the same number of candies, or (ii) Rafa gets more candies than both Lara and Emma, but Lara and Emma each receive an equal number of candies, with both receiving at least one candy each. We compute the PN and PS for the impact that Lara and Emma receiving an equal number of candies (denoted as $C_{lm}$) has on the party being 'happy' (denoted as $C_h$).

A fourth problem (`ConPref`) is included in Appendix B. The reasoning graphs for the problems `EvenSum` and `CandyParty` are shown in Figure 4 (*Right*). The structural equations corresponding to each of these graphs can be found in the Appendix C. We estimate the PN and PS for each of these problems using three difference language models: GPT-2, GPT-3.5-turbo and GPT-4 [23]. Our objective is to investigate whether the ability to reason, as conceptualised in this paper, *emerges* as the complexity and size of the models grow. While similar evaluations could be conducted using other families of LLMs, such as Llama [33], Gemini [30], Phi [11], etc., we have chosen to limit our analysis to the GPT series here for the sake of a clearer and more straightforward exposition, but more results are available in Appendix J.

To assess the reasoning abilities of various models, we use the following metrics:

1. *Factual Inconsistency Rate* (FIR): This measures the rate of inconsistencies when the models respond to factual queries.

2. *Counterfactual Inconsistency Rate* (CIR): Similar to FIR, but this metric measures inconsistencies in responses to counterfactual queries.

For a detailed explanation of these metrics, please refer to Appendix H. We estimate the standard errors of FIR and CIR by examining the variations in outputs across multiple model responses. We take this aspect into account by collecting multiple answers from the models and propagating the stochasticity of the answers to the computation of PN and PS. Additionally, we capitalise on this variability to construct the densities over the inferred PN and PS. This process involves generating 500 bootstrap samples from the model's factual and counterfactual responses[2]. From these densities, we calculate $\gamma$-PN-overlap, which measures the concentration of the probability distribution within a radius $\gamma$ around the actual PN, and $\gamma$-PS-overlap does the same for PS[3].

---

[2]Note that while generating a larger number of model answers could potentially increase accuracy, the computational costs are prohibitive. Therefore, the bootstrap approach serves as a reasonable compromise.

[3]The code to reproduce the analyses and figures can be provided upon request, and will be made open source if this work is accepted for publication.

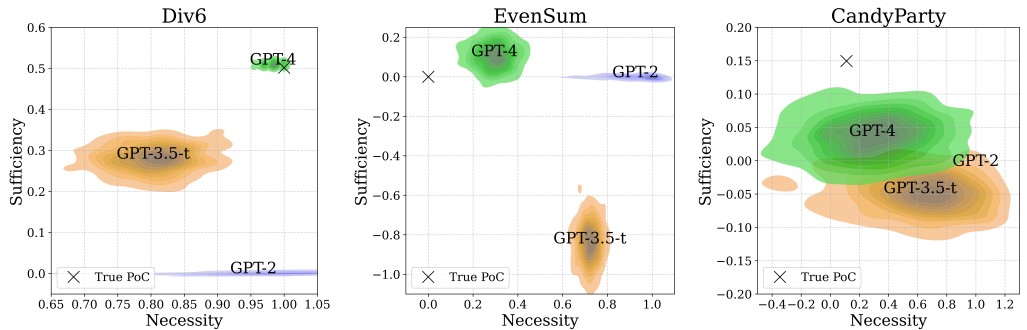

Figure 6: True PN and PS vs. inferred PN and PS using GPT-2, GPT-35-turbo and GPT-4. The densities of the estimated probabilities capture the uncertainty associated with the responses by each model.

## 4.1 Factual vs. counterfactual predictions

Figure 5 (*Left*) illustrates the alignment between the outputs of GPT-2, GPT-3.5-turbo, and GPT-4, and the factual predictions and counterfactuals for the `Div6` problem. The shading within each cell of the heatmap indicates the degree of mismatch between model-generated outputs and the true information, with the colour intensity reflecting the level of disagreement based on the 10 answers from the models. As highlighted in Figure1—where the average disagreement across the first 100 columns of these heatmaps informs the results—more sophisticated models like GPT-4 demonstrate a closer match with the counterfactuals derived from the true reasoning graph. For similar comparisons involving other problems, please refer to Appendix I.

One might wonder if the evaluation of reasoning truly requires PN and PS, or if it could be sufficiently assessed by examining only the inconsistency rates in factual/counterfactual data. Figure 5 (*Right*) underscores the importance of PN and PS. It presents the estimated distributions of PN for the `Div6` problem, based on 500 replicates under five scenarios where true counterfactuals are randomly altered with probabilities $0.005$, $0.001$, $0.05$, $0.1$ and $0.2$. As we might anticipate, the greater the deviation from a dataset free of counterfactual errors, the more significant the discrepancy from the actual $PN = 1$ for this example. Notably, even minor perturbations can lead to substantial shifts in the estimated PN. For example, with a $0.05$ probability of counterfactual perturbation, the estimated PN varies between $0.5$ and $0.9$. This suggests that relying solely on counterfactual errors could lead to an overestimation of the models' reasoning abilities, particularly their understanding of the necessary and sufficient conditions within a problem. Furthermore, a counterfactual error rate of $0.2$ in this example results in entirely inconsistent (negative) probabilities due to the mismatch between the conditional and interventional distributions, as defined in Eq. 1.

## 4.2 Evaluation of LLMs reasoning

We computed the CIR, FIR, $\gamma$-PN-overlap, and $\gamma$-PS-overlap for the problems `Div6`, `EvenSum` and `CandyParty` using GPT-2, GPT-3.5-turbo, and GPT-4.

Figure 6 illustrates the estimated PN and PS for each problem, obtained through bootstrap resampling. Each density is labeled with the model that was used used to generate the data for those results. The true values of the PS and PN in each problem is marked with a cross. A model is considered capable of reasoning if the PN-PS density estimates overlap with the true probabilities of causation. Such an overlap was only achieved by GPT-4 for `Div6` problem. Other results varied, indicating generally weak reasoning abilities. Negative values of PN and PS in several instances, are due to inconsistencies in $\mathcal{D}_F^{LLM}$ and $\mathcal{D}_{CF}^{LLM}$ as detailed in Section 4.1.

Figure 7 (*Left*, *Centre*) features the $\gamma$-PN-overlap and $\gamma$-PN-overlap curves for all models and problems, where ideal reasoning corresponds to the metrics equalling one for any value of $\gamma$. GPT-4 shows this level of reasoning for the `Div6` problem. However, GPT-2 had an accurate PN for `Even-Sum`, but the PS estimates were notably less accurate.

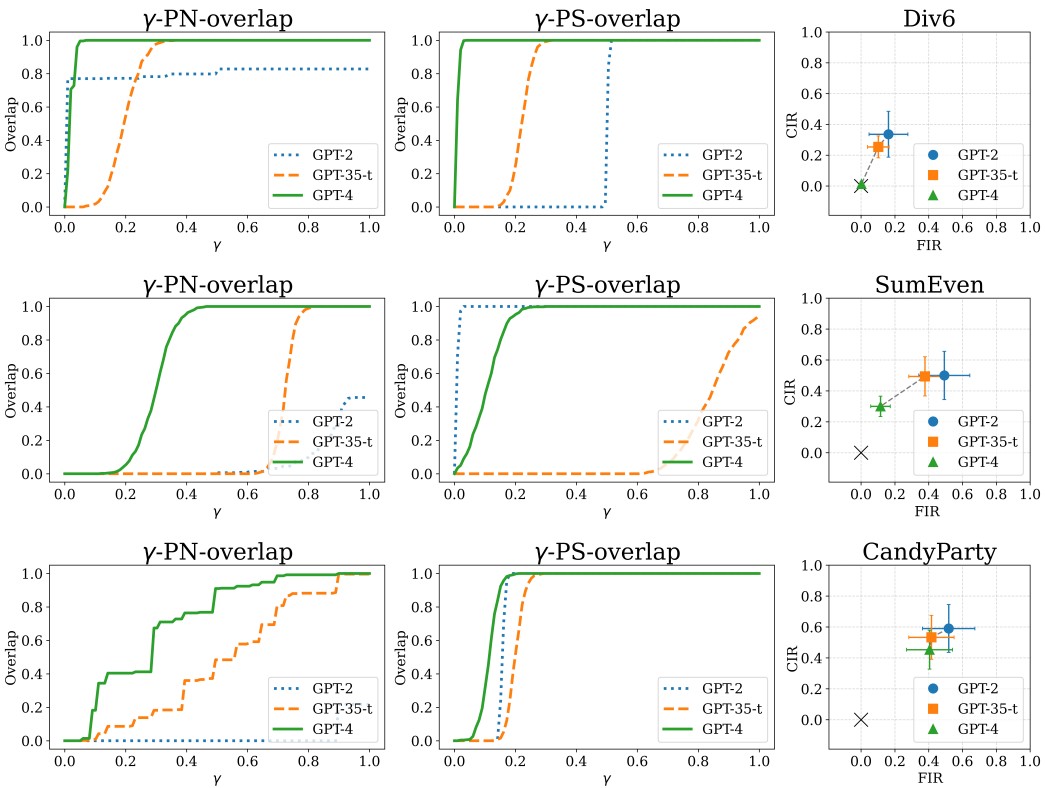

Figure 7: *Left*, *Centre*: Reconstruction of the $\gamma$-PN-overlap and $\gamma$-PS-overlap curves for GPT-2, GPT-35-turbo and GPT-4. Ideal reasoning is achieved when the overlap is one for all values of $\gamma$. *Right*: Visualization of FIR and PIR. Ideal reasoning is attained when both metrics are zero (denoted by a $\times$).

Figure 7 (*Right*) presents the values of CIR and PIR (with the standard deviations included brackets). An emerging trend towards reasoning is observed in the GPT family of models, particularly seen with GPT-4 for the `Div6` problem. An intriguing question is whether future versions of these models will similarly approach the true PN and PS for other problems as well.

## 5 Discussion

The primary objective of this paper was to explore and understand the reasoning abilities of LLMs, which is essential for their successful deployment in a range of applications. Given the growing dependence on LLMs for complex reasoning tasks, such as mathematics, programming, or strategic planning, understanding this is crucial. To evaluate these reasoning abilities, we introduced a novel framework that employs probabilistic measures of necessity and sufficiency, and find that while various models (GPT-2, GPT-3.5-turbo, and GPT-4) can replicate aspects of reasoning to some degree, they often falter when it comes to counterfactual reasoning. What makes our approach unique is that we test the models with scenarios where generalization by reasoning, rather than replicating statistical patterns in the training data, is required to provide correct answers. Notably, the ability to reason, as defined in this paper, does improve with more complex models, yet it is still far from flawless. This observation leads to the question of whether future versions of these models will achieve perfect reasoning. Our results are significant, as they reveal the limitations of LLMs, and emphasize the need for further research to enhance their reasoning capabilities.

In general, reasoning goes beyond the math examples that we have included in the paper. We believe that the same theory and tools can be used in other domains. The Hex framework, that serves as a mathematical framework to formalize our ideas, requires the definition of a query, state, and an abstract execution machine that is used to predict how the state is modified given the query. In this

framework, for instance, one could think about problems in vision, where the elements of the state are the objects in an image, and the query corresponds to a counterfactual query that describes an intervention in the environment. The concepts of necessity and sufficiency still apply in this scenario. In an image where an object is removed or altered, the framework can help determine the impact of this change on a property of the overall scene. We believe that this approach and will be key in other fields such as robotics, and or social sciences, where understanding the necessity and sufficiency between different elements is crucial for accurate reasoning and decision-making.

**Limitations**: Our approach has several limitations that we acknowledge, but did not address within the scope of this research.

1. *Dependence on reasoning graphs*: our method requires access to reasoning graphs. This requirement may hinder our ability to fully understand the reasoning abilities of LLMs in situations where it is challenging to derive relationships, including causal ones.

2. *Boolean variable restriction*: our method is designed to work with boolean valued variables, which is restrictive, particularly for cases involving multiple states or conditions occurring at the same time. However, we believe that this issue can be addresss with further research.

3. *Prompt-dependent results*: The findings we report are based on an LLM's reasoning abilities as determined by two specific types (factual/counterfacutal) of prompts that we used. Of course, other techniques like chain-of-thought prompting could be used, but our focus remains on establishing a consistent baseline. Future work could explore the impact of different prompting strategies on reasoning performance, potentially leading to more refined and effective methods for evaluating and enhancing reasoning capabilities in LLMs. Our experiments did not aim to fine-tune these prompts or to 'optimise reasoning'—a separate area of ongoing research. Instead, our goal was to offer valuable insights that can aid the community in developing new benchmarks and employing LLMs responsibly.

**Broader impact**: Evaluating the reasoning capabilities of LLMs is essential as it significantly influences their effectiveness in various domains. In education and research, it is important for the model to be able to provide accurate explanations and to formulate meaningful hypotheses. In the commercial sector, the effectiveness of automated processes/systems relies heavily on how well the model can reason. When it comes to accessibility, the model must be able to understand and meet diverse user needs, which hinges on its reasoning ability. Moreover, identifying and mitigating biases in AI systems—a key aspect of ethical and equitable AI—requires a detailed examination of the models' reasoning processes. Therefore, while LLMs hold immense promise, ensuring their responsible and beneficial use is predicated on a thorough appraisal of their reasoning abilities. We believe that our research is an important step in this direction.

## Acknowledgments

We would like to thank Ted Meeds, Alicia Curth and Sushrut Karmalkar for their invaluable discussions and feedback throughout the process of writing this paper.

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

# Appendix for: 'Does Reasoning Emerge? Examining the Probabilities of Causation in Large Language Models'

## A    Causal models, counterfactuals and probabilities of causation

This section provides technical background on structural causal models, counterfactuals and probabilities of causation [25].

**Definition 4** (Causal Model). *A causal model $\mathcal{M}$ is a triple $\langle \mathcal{V}, \mathcal{F}, \varepsilon \rangle$ where:*

1. *$\mathcal{V} = \{V_1, \ldots, V_n\}$ is a set of endogenous variables.*

2. *$\varepsilon = \{\varepsilon_1, \ldots, \varepsilon_n\}$ is a set of exogenous variables. The exogenous variables $\varepsilon$ are assumed to be independent of each other and represent the unobserved factors that influence the values of $V$.*

3. *$\mathcal{F} = \{f_1, \ldots, f_n\}$ is a set of functions. Each function $f_i$ determines the value of $V_i$ as a function of its parents $\mathrm{PA}_i \subseteq \mathcal{V} \cup \varepsilon$, where $\mathrm{PA}_i$ are the variables that directly cause $V_i$.*

Any causal model can be represented by a directed acyclic graph (DAG) $\mathcal{G}$, where the nodes represent the variables $\mathcal{V}$, and the edges are the direct causal relationships between these variables. Let $X$ be a subset of variables in $\mathcal{V}$, and $x$ be a specific realization of the values these variables can take. We define a submodel $\mathcal{M}_{X=x}$ to be a causal model $\langle \mathcal{V}, \mathcal{F}_t, \epsilon \rangle$, where $\mathcal{F}_t = \{f_i : V_i \notin T\} \cup \{X = x\}$.

**Definition 5** (Intervention, $do$ operator). *Consider a causal model $\mathcal{M} = \langle \mathcal{V}, \mathcal{F}, \varepsilon \rangle$, with $X$ being a subset of variables in $\mathcal{X}$, and $x$ a particular realization of $T$. The effect of the intervention $do(X = x)$ in $\mathcal{M}$ is given by the submodel $\mathcal{M}_{X=x}$.*

**Definition 6** (Potential outcome and counterfactual). *Let $Y$ be a variable in $\mathcal{V}$, and let $X$ be a subset of $\mathcal{V}$. The potential outcome of $Y$ resulting from the intervention $do(X = x)$, denoted by $Y_{X=x}(\epsilon) = y$, is the solution for $Y$ in the set of equations $\mathcal{F}_t$. A counterfactual is defined as the potential outcome $Y_{X=x}(\epsilon)$ for the hypothetical scenario "what would the value that $Y$ have been if $X$ had been set to $x$".*

A distribution $\mathbb{P}$ over the exogenous variables $\varepsilon$ establishes a corresponding a probability distribution over the endogenous variables $\mathcal{X}$ as well as the potential outcomes. In practical applications, $\mathbb{P}(\epsilon)$ characterizes the target population of the study. The probability of a counterfactual $Y_{X=x'}$ induced by the submodel $\mathcal{M}_{X=x}$ is:

$$\mathbb{P}(Y_{X=x} = y) = \sum_{\{\epsilon | Y_{X=x}(\epsilon)=y\}} \mathbb{P}(\epsilon)$$

In addition, probabilities of the type $\mathbb{P}(Y_{X=x'} | X = x, Y = y)$ can be computed as

$$\mathbb{P}(Y_{X=x'} = y' | X = x, Y = y) = \sum_{\epsilon} \mathbb{P}(Y_{X=x'}(\epsilon) = y')\mathbb{P}(\epsilon \mid X = x, Y = y)$$

By conditioning on $X = x$ and $Y = y$, the counterfactual outcome $y'$ under the intervention $do(X = x')$ is the expectation of the index function $Y_{X=x'}(\epsilon) = y'$ with respect to the updated probability distribution $\mathbb{P}(\epsilon | X = x, Y = y)$. Three special cases of distributions of this type are of special interest for us.

**Definition 7** (Probability of necessity, [24]). *Let $X$ and $Y$ be two binary variables in a causal model $\mathcal{M} = \langle \mathcal{X}, \mathcal{F}, \varepsilon \rangle$. The probability of necessity (PN) is defined as:*

$$PN := \mathbb{P}(Y_{X=x'} = y' | X = x, Y = y)$$

**Definition 8** (Probability of sufficiency, [24]). *Let $X$ and and $Y$ be two binary variables in a causal model $\mathcal{M} = \langle \mathcal{X}, \mathcal{F}, \varepsilon \rangle$. The probability of sufficiency (PS) is defined as:*

$$PS := \mathbb{P}(Y_{X=x} = y | X = x', Y = y')$$

The PN is the probability of observing a different outcome in the absence of the event $X = x$. The PS is the probability of $x$ to generate $y$ in cases where both had different values ($x'$ and $y'$).

**Definition 9** (Probability of necessity and sufficiency, [24])**.** *Let $X$ and and $Y$ be two binary variables in a causal model $\mathcal{M} = \langle \mathcal{X}, \mathcal{F}, \varepsilon \rangle$. The probability of necessity and sufficiency is defined as:*

$$PNS := \mathbb{P}(y_x, y'_{x'}) = \mathbb{P}(x,y)PN + P(x',y')PS$$

The PNS computes the probability that $X = x$ is the only way of obtaining $Y = y$, in other words, the probability that $X = x$ is both necessary and sufficient to observe $Y = y$. The probabilities PN, PS and PNS are not identifiable with observational or experimental data unless $Y$ is monotonic with respect to $X$, and both observational and experimental data are available [31]. If this condition is satisfied, then they are identifiable and can be computed as follows:

$$PN = \frac{\mathbb{P}(y) - \mathbb{P}(y|do(x'))}{\mathbb{P}(x,y)} \text{ and } PS = \frac{\mathbb{P}(y|do(x)) - \mathbb{P}(y)}{\mathbb{P}(x',y')}. \quad (2)$$

Note that PN and PS require the knowledge of $do(X = x)$ and $do(X = x')$. These quantities are generally unobserved for the whole population since observed individuals are only subject to one of the two conditions, unless experimental data is available.

## B ConPref problem

**Congruent preferences** (`ConPref`): Consider three real numbers $M$, $N$ and $T$. If $M \leq N$ and $N \leq T$, then $M \leq T$. We compute PN and PS for the condition $M \leq N$ ($C_{mn}$) to having enough evidence to know if $M \leq T$ ($C_{mnt}$). If $M \leq N$ or $N \leq T$ are false then $C_{mnt}$ is false. For our evaluation, we consider all combinations of values for $M$, $N$ and $T$, for numbers between 1 and 8.

**Reasoning graph**

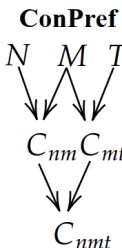

Figure 8: Reasoning graph for the `ConPref` problem.

**Reasoning results**

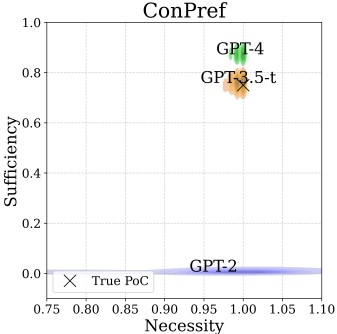

Figure 9: True PN and PS vs. inferred PN and PS using GPT-2, GPT-35-turbo and GPT-4 for the `ConPref` problem. .

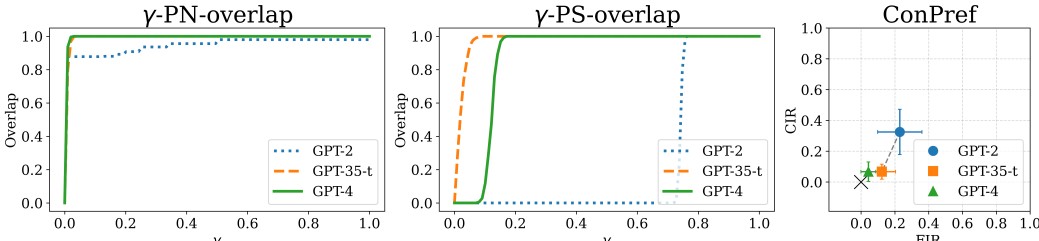

Figure 10: *Left, Centre*: Reconstruction of the $\gamma$-PN-overlap and $\gamma$-PS-overlap curves for GPT-2, GPT-35-turbo and GPT-4 for the `ConPref` problem. Ideal reasoning is achieved when the overlap is one for all values of $\gamma$. *Right*: Visualization of FIR and PIR. Ideal reasoning is attained when both metrics are zero (denoted by a $\times$).

## C  Problems: structural equations

### C.1  `Div6`

$$
\begin{aligned}
N &\sim \mathbb{P}_N \\
C_2 &= N \pmod 3 \equiv 0 \\
C_3 &= N \pmod 2 \equiv 0 \\
C_6 &= C_2 \wedge C_3
\end{aligned}
$$

where $C_2$, $C_3$ and $C_6$ represent boolean values that indicate whether the number is divisible by 2, 3 and 6 respectively. $\mathbb{P}_N$ is the mechanism for generating the original numbers (1 to 400 in our examples).

### C.2  `EvenSum`

$$
\begin{aligned}
N &\sim \mathbb{P}_N \\
M &\sim \mathbb{P}_M \\
T &\sim \mathbb{P}_T \\
C_n &= N \pmod 2 \\
C_m &= M \pmod 2 \\
C_t &= T \pmod 2 \\
C_{nmt} &= (C_n + C_m + C_t = 1) \wedge (C_n + C_m + C_t = 3)
\end{aligned}
$$

where $C_n$, $C_m$, $C_t$ and $C_{nmt}$ represent boolean values and $\mathbb{P}_N$, $\mathbb{P}_M$, $\mathbb{P}_T$ are the mechanisms to generate the original numbers (1-8 in our examples).

### C.3  `ConPref`

$$
\begin{aligned}
N &\sim \mathbb{P}_N \\
M &\sim \mathbb{P}_M \\
T &\sim \mathbb{P}_T \\
C_{nm} &= N \leq M \\
C_{mt} &= M \leq T \\
C_{nmt} &= C_{nm} \wedge C_{mt}
\end{aligned}
$$

where $C_{nm}$, $C_{mt}$ and $C_{nmt}$ represent boolean values and $\mathbb{P}_N$, $\mathbb{P}_M$, $\mathbb{P}_T$ are the mechanisms to generate the original numbers (1-8 in our examples).

$$
\begin{aligned}
R &\sim \mathbb{P}_R \\
L &\sim \mathbb{P}_L \\
E &\sim \mathbb{P}_E \\
C_{r>0} &= R > 0 \\
C_{l>0} &= L > 0 \\
C_{e>0} &= E > 0 \\
C_{r\geq 2} &= R \geq 2 \\
C_{l\geq 2} &= L \geq 2 \\
C_{e\geq 2} &= E \geq 2 \\
C_{rl} &= R > L \\
C_{re} &= R > E \\
C_{l=e} &= L = E \\
C_{r\geq 0, l\geq 0, e\geq 0} &= C_{r\geq 0} \wedge C_{l>0} \wedge C_{e\geq 0} \\
C_{r>l, r>e} &= C_{rl} \wedge C_{re} \\
C_{r\geq 2, l\geq 2, e\geq 2} &= C_{r\geq 2} \wedge C_{l\geq 2} \wedge C_{e\geq 2} \\
C_h &= (C_{r\geq 2, l\geq 2, e\geq 2} = 1) \wedge (C_{l=e} \vee C_{r\geq 0, l\geq 0, e\geq 0})
\end{aligned}
$$

where $\mathbb{P}_R, \mathbb{P}_L, \mathbb{P}_E$ are the mechanisms to generate the original numbers (all combinations in which 20 candies can be shared in our example).

# D  Proof of LLMs Zero-counterfactual consistency

*Proof.* Commutability of the Hex diagram implies that all paths from $\sigma_0$ to $\sigma_1$ result in the same outcome. This holds for all counterfactuals, which implies that $Y_{X=x}^{LLM} = Y_{X=x}$ for any value of $X$ and $Z$. Therefore:

$$
\mathbb{E}_{\mathbb{P}(X,Y,Z)}\left[Y_{X=x}^{LLM} \neq Y_{X=x}\right] = 0
$$

for any $\mathbb{P}(X,Y,Z)$. $\qquad\square$

# E  Direct and counterfactual prompts

## E.1  `Div6` **problem**

**Direct Prompt:**

"Does 6 divide {'X'}?  Use the factor method to answer this question.  Be as concise as possible."

**Counterfactual Prompt:**

"Imagine that {'X'} {'has'/'has not'} 3 as prime factor while retaining all its other prime factors.  With this assumption does {self.divisor} divide {'X'}?  Use the factor method to answer this question.  Be as concise as possible."

## E.2  `EvenSum` **problem**

**Counterfactual Prompt:**

"Let N, M and T be three integers.  Then N+M+T is even if the three numbers are even or if only one is even and the remaining two are odd.  Consider the numbers N={N}, M={M} and T={T}.  Is N+M+T even?  Be as concise as possible."

**Direct Prompt:**

Let N, M and T be three integers.  Then N+M+T is even if the three numbers are even or if only one is even and the remaining two are odd.  Consider the numbers N={N}, M={M} and T={T} and imagine that N {is/is not} even. With this assumption, is N+M+T even?  Be as concise as possible."

### E.3 `ConPref` **problem**

**Direct Prompt:**

"Let N, M and T be three integers.  We know that if N is smaller or equal that M and M is smaller or equal than T then N is smaller or equal than T.Consider the numbers N={N}, M={M} and T={T}.  By only looking at the relationships (N={N} vs.  M={M}) and (M={M} vs.  T={T}), can we know if N is smaller or equal that T? Be as concise as possible."

**Counterfactual Prompt:**

"Let N, M and T be three integers.  We know that if N is smaller or equal that M and M is smaller or equal than T then N is smaller or equal than T. Consider the numbers N={N}, M={M} and T={T}.  Now imagine that the number N {'is smaller or equal '/ 'is not smaller or equal'} than M. Even if this contradict the values of the numbers X and Y, use this assumption and the relationships between and M={M} and T={T}, to decide if can we tell if N is smaller or equal that T? Don't make any conclusion or comment based on the values, just based on the assumption and the relationships.  Be as concise as possible."

### E.4 `CandyParty` **problem**

**Direct Prompt:**

"Rafa has invited Lara and Emma to his birthday party.  He has {num candies} to distribute among them.  They all will be happy in the party in one of the following cases:  1) Each of them gets at least 2 candies or 2) Lara and Emma get the same number of candies, but at least one candy each, and Rafa gets more than them.  After distributing the candies, Lara gets {L}, Emma gets {E} and Raphael gets {R} candies.  With this candies distribution, will they all be happy in the party?  Be as concise as possible."

**Counterfactual Prompt:**

Rafa has invited Lara and Emma to his birthday party.  He has {num candies} candies to distribute among them.  They all will be happy in the party in one of the following cases:  1) Each of them gets at least 2 candies or 2) Lara and Emma get the same number of candies, but at least one candy each, and Rafa gets more than them After distributing the candies.  After distributing the candies, Lara gets {L}, Emma gets {E} and Rafa gets {R} candies.  Consider the number of candies distributed to each of them and imagine that they think that {'Lara and Emma have the same number of candies'/'Lara and Emma have different number of candies'}.  With this assumption, will they all be happy in the party?  Be as concise as possible."

## F GPT-4 concretization

**Prompt:** `"You are an entity extractor expert. I am going to give you`
`a question-answer pair. I want you to say if the meaning of answer is`
`positive or negative. If the answer has words like 'Yes' this will make`
`it positive. If the answer contains words like 'No' this will make it`
`negative. Always answer with only one word (Positive or Negative). For`
`the question '{question}' and the answer '{answer}' the meaning is"`

## G HEX for counterfactual query

$$
\begin{array}{ccc}
\hat{\sigma}_0 & \xrightarrow{\quad Q_{C_6} \quad} & \hat{\sigma}_1 \\
\uparrow {\scriptstyle \alpha_c} & & \downarrow {\scriptstyle \gamma} \\
\end{array}
$$

$$
\sigma_0 = \begin{cases} N \mapsto 10, \\ C_3 \mapsto F, \\ C_6 \mapsto F, \\ C_2 \mapsto T \end{cases} \xrightarrow{\quad Q_{C_3} = True \quad} \sigma_0' = \begin{cases} N \mapsto 10, \\ C_3 \mapsto T, \\ C_6 \mapsto F, \\ C_2 \mapsto T \end{cases} \xdashrightarrow{\quad Q_{C_6} \quad} \sigma_1 = \begin{cases} N \mapsto 10, \\ C_3 \mapsto T, \\ C_6 \mapsto T, \\ C_2 \mapsto T \end{cases}
$$

Figure 11: The HEX diagram for a counterfactual query in the `Div6` problem. We split the query in two sub-queries $Q_{C_3=True}$ and $Q_{C_6}$ that performs the two operations need to compute the counterfactual state. $Q_{C_3=True}$ only sets the value of $C_3$ to True. $Q_{C_6}$ replaces the value of $C_6$ by its counterfactual. This operation can be executed via the concrete path (using the structural causal model of the problem) of by using an LLM.

## H Evaluation metrics

Let $n$ be the number of instances of each problem. For example, $n = 400$ for the `Div6` problem because we use the first $400$ integers to test reasoning. For the intervention node $X$ and the outcome note $Y$, we distinguish between factual predictions $Y|X$ (simulated from the original reasoning graph) and counterfactual predictions $Y_{X=x}$ (simulated from the intervened graph). We respectively denote the LLM versions of this quantities as $Y^{LLM}|X = x$ and $Y^{LLM}_{X=x}$, which are computed via factual and counterfactual prompts.

$$\text{FIR} := \frac{1}{n} \sum_{i=1}^{n} \mathbb{I}\left[(Y^{LLM}|X = x) \neq (Y|X = x)\right]. \tag{3}$$

$$\text{CIR} := \frac{1}{n} \sum_{i=1}^{n} \mathbb{I}\left[Y^{LLM}_{X=x} \neq Y_{X=x}\right], \tag{4}$$

Let $m$ be the number of bootstrap samples used from the binary answers of the LLM. $\hat{PN}_i$ and $\hat{PS}_i$ are estimation of $PN$ and $PS$ for the ith bootstrap sample. Then

$$\gamma - \text{PNO} := \frac{1}{m} \sum_{j=1}^{m} \mathbb{I}\left[|\hat{PN}_i^{LLM} - PN| \leq \gamma\right] \tag{5}$$

$$\gamma - \text{PNS} := \frac{1}{m} \sum_{j=1}^{m} \mathbb{I}\left[|\hat{PS}_i^{LLM} - PS| \leq \gamma\right] \tag{6}$$

# I    Element-wise PIR and FIR

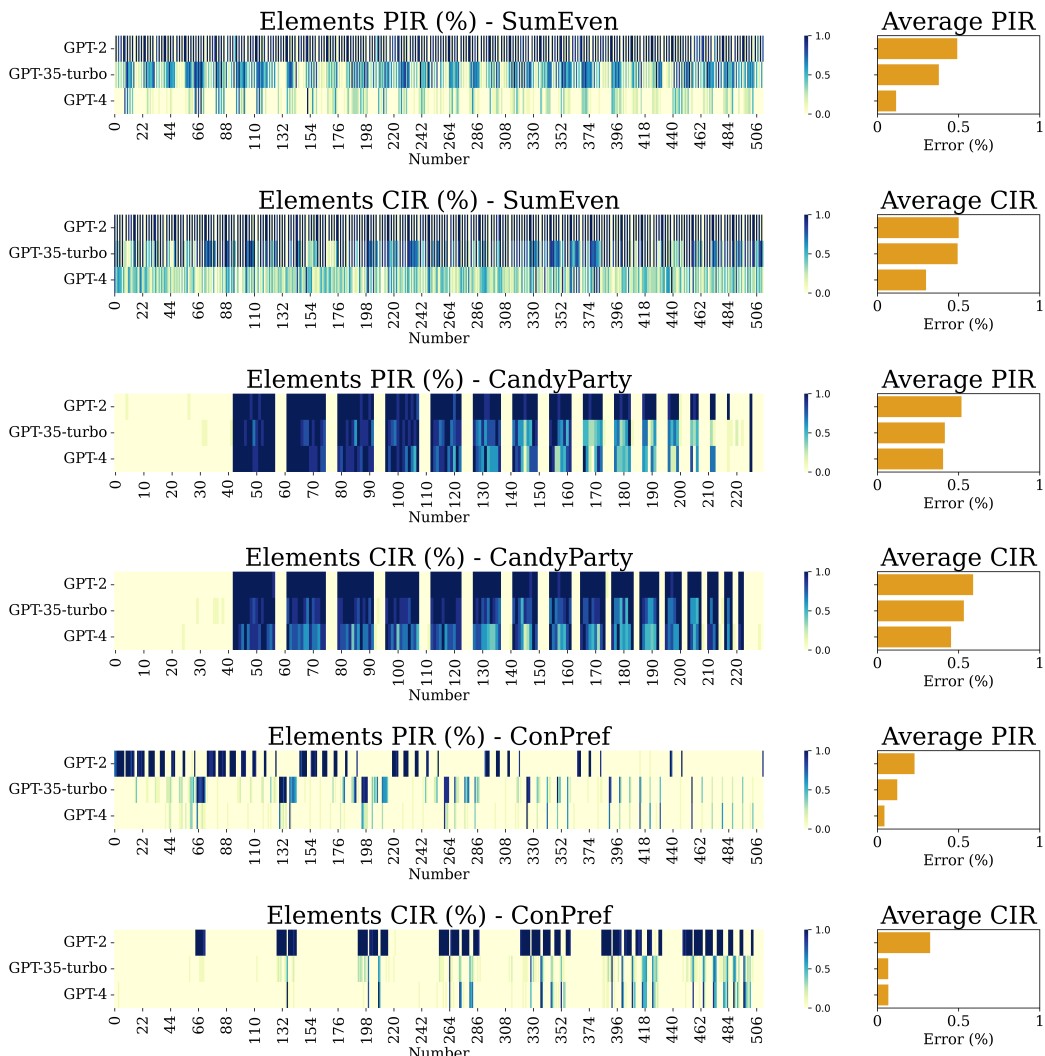

Figure 12: Element and aggregated CIR and FiR for the SumEven and CandyParty problems.

# J   Experiments with other model families

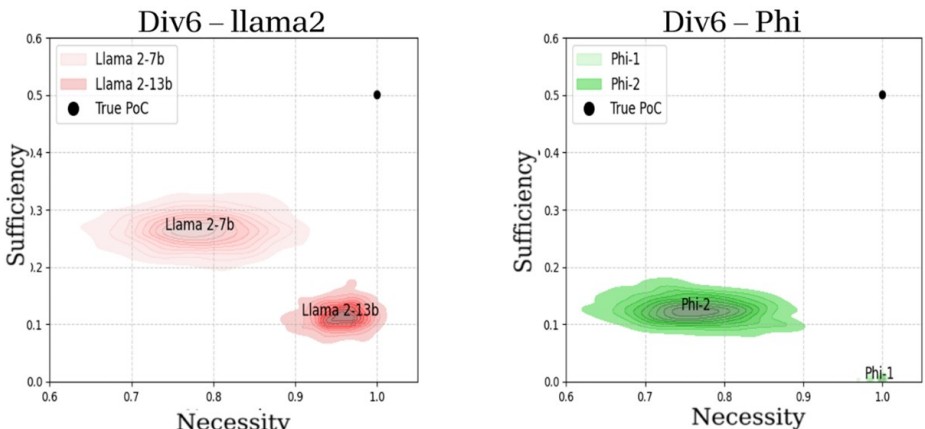

Figure 13: in other families of models. We re-run the Div6 problem with the same setup used in the paper with two new families of models: Llama (7-7b and 13b) and Phi (1,2). The results are consistent with the findings discussed in the main body of the paper

