# OpenReview forum: "Does Reasoning Emerge? Examining the Probabilities of Causation in Large Language Models"
_NeurIPS.cc/2024/Conference — NeurIPS 2024 poster_

### Official Review · Reviewer_Hvct · 2024-06-28

**Soundness:** 3
**Presentation:** 3
**Contribution:** 2
**Rating:** 4
**Confidence:** 3

**Summary:**

This paper investigates the reasoning capabilities of large language models (LLMs) by examining the concepts of necessity and sufficiency, which are key elements of logical reasoning. To assess the LLMs' reasoning abilities, the authors introduce a framework that computes the probability of necessity (PN) and the probability of sufficiency (PS) by comparing the actual values derived from factual and counterfactual datasets with those simulated by the LLMs. The paper presents a reasoning test for assessing LLMs' reasoning abilities using a divisibility problem as an example. The authors create factual and counterfactual datasets based on a reasoning graph and compare the actual PN and PS values with those generated by the LLMs. The results show that the closer the estimated PN/PS values are to the actual PN/PS values, the better the LLM is at reasoning.

**Strengths:**

The main strength of the paper is the introduction of a systematic method for evaluating the reasoning capabilities of large language models (LLMs) by examining the concepts of necessity and sufficiency. This novel framework can be beneficial for researchers and developers working on improving LLMs. The method tries to evaluate from two different angles:  the accuracy with which an LLM solves a problem, and its capacity to understand and process the fundamental elements that lead to that solution. I think the authors tackles an important problem in LLM reasoning, which is an important and ongoing debate about the extent to which LLMs are capable of actual reasoning.

**Weaknesses:**

- The paper uses a single example, the divisibility problem, to demonstrate the reasoning test. While this example helps illustrate the concepts, it may not be representative of the wide range of problems LLMs are expected to solve. Including additional examples from various domains would strengthen the paper's claims and generalizability.
- The paper does experiments on specific LLMs (gpt series) but doesn't provide results on other models or comparisons between multiple models. Including a variety of LLMs with different architectures and training datasets would provide a more comprehensive evaluation of the proposed framework. And, as GPT series are not open-sourced, it's unclear how OpenAI's hidden prompts or value-alignment finetuning will influence the results.
- The paper focuses on the probabilistic interpretations of necessity and sufficiency in a creatain math problem but does not explore a wider range forms of reasoning (I'm not seeing this point as a signifigant weaknesses though, but I think the word "reasoning" actually covers more complex scenarios).

Also see the questions below.

**Questions:**

- How do the authors distinguish the results from the influence of the LLMs' prior knowledge? The LLMs' performance in the reasoning test may be affected by the patterns and relationships they have learned from their training data, especially commonsense reasoning questions related to Math. Are there any steps taken to control or account for this influence when evaluating the LLMs' reasoning abilities? In the interpretation of results, how do you account for the potential influence of the prior knowledge and the LLM's understanding of reasoning capabilities?
- Have the authors tried any prompt tuning or do they have some primary resuls? Although the authors have made this point clear in the limitations, I think the prompt actually greatly influence the output of the LLMs and may have a influence on the conclusions.

---

> ### Author Rebuttal · Authors · 2024-08-06
>
> Thank you for highlighting the strengths of our paper. We appreciate your recognition of our systematic method for evaluating LLM reasoning through the probabilities of necessity and sufficiency. Your positive feedback on our dual-angle approach comparing the predictive vs the reasoning abilities of LLMs and its relevance to the ongoing debate about LLM reasoning is encouraging.
>
> **Comment of weaknesses:**
>
> *1. On single example:* While we agree that assessing reasoning skills through additional tasks would be beneficial, such exploration exceeds the intended scope of this work. The primary objective of this paper is to describe and demonstrate the significance of the probabilities of causation as essential metrics for evaluating reasoning in large language models. The development of comprehensive benchmarks for reasoning is indeed the focus of our current research.
>
> *2. Regarding experiments with open-source models:* Our paper concentrates on the GPT series to make our exposition clearer but the same principles are relevant to other open-source models as well. We have added further results on the Div6 issue in the extra page, utilizing two more model families—Phi (1, 2) and Llama (2-7b, 2-13b)—which show consistent behaviour with the findings discussed in the paper. These figures have also been included in the appendix of the paper.
>
> *3. Other reasoning tasks:* We agree that reasoning, in general, goes beyond the math examples that we have included in the paper. We believe that the same theory and tools can be used in other domains, and this will be the focus of our future research. The Hex framework, that serves as a mathematical framework to formalize our ideas, requires the definition of a query, state, and an abstract execution machine that is used to predict how the state is modified given the query. In this framework, for instance, one could think about problems in vision, where the elements of the state are the objects in an image, and the query corresponds to a counterfactual query that describes an intervention in the environment. The concepts of necessity and sufficiency still apply in this scenario. In an image where an object is removed or altered, the framework can help determine the impact of this change on a property of the overall scene. This approach can be extended to other fields such as robotics, and or social sciences, where understanding the necessity and sufficiency between different elements is crucial for accurate reasoning and decision-making. We will further clarify this point in the paper to make sure the readers are aware of the generality of our approach.
>
> **Answers to questions:**
>
> *1. Influences of LLM previous knowledge:* As illustrated in Figure 1, a key aspect of our approach is to make a clear distinction between answers that the model is giving based on likely previously collected knowledge (factuals) and based on scenarios in which it is very unlikely that the model has been trained on (counterfactuals). Using the Div6 example, one can expect that the concepts of divisibility and the factor method are present in the training data set of a language model. Indeed, in Figure 1, the three models achieve a low error rate on this question, with GPT-35-turbo achieving errors close to zero. However, it is less likely to expect that imaginary scenarios where the standard rules of arithmetic have been violated to be present in the training data. This is consistent with Figure 1 and all the results in the experimental section - the counterfactual errors are significantly larger than factual errors for all models and problems. What makes our approach unique is that we test the models with scenarios where generalization by reasoning, rather than replicating statistical patterns in the training data, is required to provide correct answers. We have clarified this point in the main body of the paper.
>
> *2. On prompt tuning.* We agree that prompt tuning may affect the results. However, as we detail in the limitations of the work, it is not our aim to optimize reasoning in LLMs, but rather to propose metrics that can characterize whether reasoning is happening. To guarantee fairness across all comparisons in the paper, we have used the same patterns in the way we write the prompts across all experiments (in a zero-shot manner). Of course, other techniques like chain-of-thought prompting could be used, but our focus remains on establishing a consistent baseline. Future work could explore the impact of different prompting strategies on reasoning performance, potentially leading to more refined and effective methods for evaluating and enhancing reasoning capabilities in LLMs. We have expanded the discussion of this topic in the discussion in a new version of the paper that we will make available upon acceptance.

---

### Official Review · Reviewer_Hhfy · 2024-07-11

**Soundness:** 3
**Presentation:** 3
**Contribution:** 3
**Rating:** 6
**Confidence:** 3

**Summary:**

To assess the reasoning abilities of large language models (LLMs) in complex tasks e.g., causation scenarios, this paper introduces a novel framework that utilizes probabilistic measures of necessity (PN) and sufficiency (PS). Through a series of mathematical examples, the study computes approximations of PN and PS using reasoning graphs and LLM that generate factual and counterfactual datasets. The generated results from the LLMs are then compared with true simulated PN and PS values from reasoning graphs. The experimental findings indicate an emerging trend towards improved reasoning capabilities within the GPT family of models, including GPT-2, GPT-3.5-turbo, and GPT-4.

**Strengths:**

+ The paper provides a clear example case at the beginning, effectively illustrating the differences in reasoning abilities among various GPT models.
+ The examination of the concepts of necessity and sufficiency is well-founded, as these are crucial elements in causation and logical reasoning tasks, making them suitable for measuring LLMs' reasoning abilities.
+ The experimental design and results are easy to understand, facilitating comprehension of the study's findings.

**Weaknesses:**

- Does four mathematic experiments representative in causal reasoning tasks?
- Before presenting Sections 2 and 3 on the probabilities of causation in an LLM, the paper should first explain the method for computing true PS and PN values using reasoning graphs to provide better context for readers.
- In Section 4.2 line 261: “Each density is labeled with the model that was used used to generate xxx”->typos

**Questions:**

It would be clearer if the author(s) could explain:
- What is the relationship between the experiment evaluation metrics, Factual Inconsistency Rate (FIR) and Counterfactual Inconsistency Rate (CIR), and the results of PN and PS?
- Could you explain more about how the datasets (factual and counterfactual) were acquired from reasoning graphs?
- What does the random noise introduced in the true counterfactuals mean, and how did you measure the noise level?

**Limitations:**

The limitations and broader impact of the study are clearly outlined in the paper.

---

> ### Author Rebuttal · Authors · 2024-08-06
>
> We thank the reviewer for the insightful comments and positive feedback. We appreciate your acknowledgment that our exposition is clear and that the necessity and sufficiency are crucial elements in reasoning. We’re also pleased that you found our experimental design and results easy to understand. Your feedback is valuable and will help enhance our work.
>
> **Comment of weaknesses:**
>
> *1. Maths and reasoning tasks*: we would like to remark here that the goal of our paper is not to evaluate causal reasoning in LLMs. Our aim is to evaluate reasoning in a general sense, and we find that some core ideas from the causality literature (the probabilities of causation) are a natural way to evaluate that. Indeed, mathematical statements like the one used in the paper are deterministic, but testing if a probabilistic system (like an LLM) can reproduce them accurately is an inference problem. By no means are we implying that the arithmetic examples in the paper represent any sort of benchmark for causal reasoning for LLMs. Instead, the reason why we used arithmetic examples is because their logic is unambiguous and easily testable, so they provide a useful basis for evaluating reasoning in LLMs and developing the concepts presented in our work.
>
> *2. Order in exposition:* we appreciate the suggestion to change the order in which the computation of PN and PS is presented, which we have considered in an updated version of our work.
>
> *3. Typo:* Thank you for catching up the typo. We have now been corrected it in the paper.
>
> **Answers to questions**:
>
> *1.Connection between CIR, FIR, PN and PS*: We appreciate this comment. We have clarified this important connection in a new version of the paper, that will be made available upon acceptance.  To analyse it in more detail, let’s start with the eight frequencies that we need to compute such metrics. First, the 4 factual frequencies: (y, x), (y’, x’), (y’, x) and (y, x’). Second the 4 counterfactual ones (y, do(x)), (y’, do(x’)), (y’, do(x)) and (y, do(x’)). The FIR and CIR are the average factual and counterfactual rates. The computation of an exact FIR requires all factuals, while. an exact CIR requires all counterfactuals. In the computation of PN and PS however, each quantity requires some factuals and counterfactuals to be correct that we now clarify. To compute PN we need to compute P(y), P(y, x) and P(y|do(x’)). This means that only the frequencies of (y, x), (y, x’) and (y, do(x’)), (y’, do(x’)) are needed. A model with perfect error rate in these quantities will approximate the PN perfectly irrespectively of the accuracy of the rest of factual and counterfactuals. On the other hand, to estimate the PS we need to compute P(y), P(y, x’) and P(y|do(x)). This requires to correctly compute the set of frequencies (y, x), (y, x’) (y, x’) and (y, do(x)), (y’, do(x)), which are different to the case of the PN. This is the reason why in some of the experiments we see small models like GPT2 doing a good job approximating the PN (even better that other more powerful modes) but a terrible job approximation of the PS. Even with low FIR and CIR values, the results of what factual and counterfactuals are correct (all the positives or all the negatives) creates this imbalance. In summary, each one of the four metrics CIR, FIR, PS and PS, requires of a different set of factual and counterfactuals to be estimated correctly. However, the pairs of metrics CIR-FIR and PN-PS requires correctness in all factual and counterfactuals.
>
> *2. Details about data generation:* True factual data in all the examples are generated by following the structural equation models included in Appendix C. The set of numbers to use in each example is detailed in the experimental section. For example, in the Div6 example (appendix C.1), we start with all the numbers between 1 and 400. We compute their modulo 3 and 2 (variables C2 and C3) and we multiply these vectors to compute C6. Counterfactual data are generated by the same generative process but replacing the value of the variable we are intervening on by the intervention value.
>
> *3. Random noise in true counterfactuals?* In our work we do not add any noise to the counterfactual explicitly. The noise is the result of extracting multiple replicates from each query to the LLM.

---

> > ### Comment · Reviewer_Hhfy · 2024-08-13
> >
> > Thanks to the authors for their responses. I appreciate the effort they’ve put into addressing the feedback. The clarifications help enhance my understanding of the work.

---

### Official Review · Reviewer_65be · 2024-07-12

**Soundness:** 2
**Presentation:** 2
**Contribution:** 2
**Rating:** 4
**Confidence:** 3

**Summary:**

The paper introduces a systematic method for assessing the reasoning capabilities of large language models (LLMs) by focusing on the concepts of necessity and sufficiency in logical reasoning. It leverages a probabilistic interpretation of these concepts and uses a reasoning graph based on boolean conditions to test the models.

**Strengths:**

1. The paper introduces a novel method for evaluating the reasoning capabilities of LLMs, which  which is a critical issue in the field of LLMs.

2.The paper includes numerous diagrams, effectively illustrating the background and contributions of the study.

**Weaknesses:**

1. The workload and the introduction of related work in the main body of the article appears somewhat insufficient. It is recommended to incorporate the more significant content from the appendix into the main text.

2. Due to the lack of more rigorous and in-depth theoretical analysis, the results may not have sufficient theoretical persuasiveness. Thus, the paper would be more valuable if it provided some conclusive summaries or proposed improvement methods for the assessment results.

**Questions:**

1. Is subsection 3.2 of the paper complete?

2. The paper treats large language models (LLMs) as black boxes to measure the probabilities of causation. Does this approach have deeper significance and value? Additionally, is this evaluation method applicable to recurrent neural networks (RNNs) or large vision models (LVMs)?

**Limitations:**

The limitations are mentioned by the author in Section 5.

---

> ### Author Rebuttal · Authors · 2024-08-06
>
> We thank the reviewer for the appreciation that the work proposes a novel method for a critical issue and the diagrams help to communicate our ideas. We appreciate the feedback, that we have now incorporated in the manuscript.
>
> **Comment of weaknesses:**
>
> *1. Incorporating more content in the main text:* Although we would have like to include more content in the main body of the paper the page limitation of the submission format forced us to leave some content in the appendix. If the paper is accepted, we will take this consideration into account since an extra page is permitted.
>
> *2. On conclusive summaries or proposed improvement:* We appreciate this comment, and we have incorporated this summary in the introduction of an updated version of our work that we will make available upon acceptance.
>
> **Answers to questions:**
>
> *1. Completeness of Section 3.2*: The section 3.2 is indeed competed, with some more relevant material in the appendix. However, to make the transition smoother to the next section we have added a last concluding sentence that explains the motivation of Lemma 1 in an updated version of the paper.
>
> *2. Applicability of this approach to other models:* Although in our work we focus on language, the same theory and tools can be used in other domains, and this will be the focus of our future research. The Hex framework, our core mathematical structure for conceptualizing these theories, requires defining a query, state, and an abstract computational engine that forecasts state alterations in response to the query. For example, within this paradigm, one might address challenges in computer vision: the 'state' comprises various objects in a visual setting, while a 'query' might represent a hypothetical modification to that environment. The principles of necessity and sufficiency are pertinent here as well. If an object in an image is edited or excluded, our framework could assess the resulting impact on a particular characteristic of the scene. This technique is equally applicable other disciplines where discerning the essential and causal relationships among components is critical for sound analysis and decision processes. It can also be applied to other models (like RNNs and LVMs) in which both factual and counterfactuals queries can be encoded.

---

### Official Review · Reviewer_1YAi · 2024-07-12

**Soundness:** 3
**Presentation:** 3
**Contribution:** 2
**Rating:** 6
**Confidence:** 3

**Summary:**

This paper evaluates the reasoning capabilities of LLMs within the framework of Judea Pearl's hierarchy of causality, focusing particularly on the ability to perform counterfactual reasoning.

LLMs are perceived as (non-deterministic) abstract machines within the HEX framework. The study assesses these models by comparing their performance on factual versus counterfactual problems, which are structured as reasoning graphs of boolean conditions. The necessity (PN) and sufficiency (PS) probabilities are estimated for 4 reasoning tasks: Divisibility by 6, Even Sum, Candy Party, and ConPref, ranging in complexity.

The paper investigates whether the actual PN and PS align with those estimated from both factual and counterfactual data, using models from the GPT family (GPT-2, GPT-3.5, and GPT-4).

Findings show limited reasoning capabilities for the evaluated model family, with GPT-4 performing the best on the divisibility task.

**Strengths:**

- Applies probabilistic measures of necessity and sufficiency to asses specifically the reasoning capabilities of LLMs, offering a nuanced evaluation that extends beyond the accuracy metric.

- Evaluates and compares LLMs performance on both factual and counterfactual problems.

**Weaknesses:**

- From the introduction "...reasoning is typically understood to be the ability of these models to demonstrate emergent capabilities that surpass mere statistical pattern recognition in the training set.". Without knowledge of or control on pre-training/post-training data for the GPT model family, it is difficult to infer that a measured reasoning capability via the introduced framework is due to "emergence" and not pattern recognition in the training data.

- The framework's applicability is reduced to reasoning tasks that can be represented as graphs of boolean variables, which may limit its broader adoption.

- Line 84-85: "The closer the estimated PN/PS values to the actual PN/PS values, the better it is at reasoning. " needs further elaboration. Specifically, an explanation of how these metrics correlate with enhanced reasoning capabilities would help in understanding the effectiveness of the framework.

- The framework imposes that LMs are abstract machines, suggesting deterministic behavior. Yet estimating probabilities of necessity (PN) and sufficiency (PS) inherently relies on variability in model responses.

- Statistical significance of observed results could further support the validity of the framework.

**Questions:**

- I assume all findings are based on temperature > 0 to fulfill the variability criterion.
Did you generate multiple completions for each prompt (factual and counterfactual), and if so, how were these incorporated into the final analysis? "10 replicated tests" is mentioned under Figure 5, How these 10 contribute to the final findings?

- Was prompt optimization attempted for smaller models as well, or just for GPT-4? Given the sensitivity of LMs to prompts, could there be a bias towards GPT-4 because it handles more complex prompts better?

-While the paper discusses PN and PS as measures of reasoning, there is less focus on how these measures help in interpreting the decisions making process. What are the main benefits in comparison to just using accuracy?

**Limitations:**

yes. See Weaknesses and Questions.

---

> ### Author Rebuttal · Authors · 2024-08-06
>
> We thank the reviewer for the feedback and appreciate the comment that our methods offers an nuanced evaluation that extends beyond the use of accuracy to evaluate reasoning in language models metric.
>
> **Comment of weaknesses:**
>
> *1. Emergence vs pattern recognition:* As illustrated in Figure 1, our method makes an explicit differentiation between answers given from probable pre-collected knowledge (factuals) and those from situations where the model likely hasn't been trained (counterfactuals). While it is likely that that an LLM was trained on factual examples (like the ones in the Div6 problem) it is very unlikely that counterfactuals containing imaginary scenarios were used in the training data set. To reinforce this point, we have included an extra experiment in the extra page allowed in this rebuttal. In the CandyParty problem, we use synthetically generated data to fine tune Phi-3-mini-128k-instruct" using counterfactual data. We observe that when counterfactual examples are used to fine tune the model (for the node "L=E"), the approximations to the true PN and PS drastically improve. In addition, we observe that the approximation of the PN and PS for other nodes in the graph also improves ("R>E,R>L") which shows generalization abilities in counterfactual scenarios. We will include this result in the experimental section of the paper if this work is accepted.
>
> *2. Boolean nodes:*  as we have acknowledged in the discussion of our work, the use of binary variables may represent a limitation. However, we believe that solutions are possible.  For example, it is possible to add dummy nodes in the graph by defining new binary variables using a threshold, and therefore condition the computation of PN and PS to such threshold. This will lead to two PN and PS probability curves for different values of the threshold that can be used to evaluate reasoning with continuous nodes.
>
> *3. PN, PS approximation and emergence:*  we appreciate the comment, and we have clarified this point in an update version of the the paper. In our results for the GPT family of models, we observe that the more sophisticate the model is (GPT4 > GPT35 > GPT2) the lower are the errors in approximating factual, counterfactuals, PN and PS.  This indeed correlates with the common knowledge of other emergent properties in this family. The metrics that we propose in our work are therefore in alignment with the patterns observed in the literature. However, they highlight that if we define reasoning as the ability of a language to replicate necessity and sufficiency, there is still room from improvement even for basic arithmetic examples.
>
> *4. LMs as abstract machines, deterministic behaviour:* we would like to clarify that although we consider LLMs as abstract execution machines, we don’t imply that they are deterministic. Indeed, we believe that this is an important and relevant aspect of our work. Because LLMs are not deterministic (the same prompt may provide multiple answers) validations tools need to have a statistical nature, and this is indeed the approach that we take in our work. In the experimental section, we take this aspect into account by collecting multiple answers from the models and propagating the stochasticity of the answers to the computation of PN and PS. We have clarified this point in the main body of the paper.
>
> *Statistical significance:*  Although individual results for statistical tests can be included, the gamma-overlap in the experimental section captures the concentration of the probability distribution within a radius γ around the true PN, PS. We followed this approach, rather than providing a single statistical test for a given significance to better illustrate the behaviour of the metrics and their trade-offs. In the case of the FIR and CIR, we have included confidence intervals in all the results, which can also be used to test the statistical validity of the results.  See Figure 7 for details.
>
> **Answers to questions:**
>
> *1. Temperature and replicates:* In the all the experiments, we kept the default temperature (temp = 1) in all the models. For each factual and counterfactual questions we collected 10 answers that we later bootstrapped (500 times) to build the full distribution over PN and PS as described in Figure 2.  The densities of PN and PS in Figure 6 are the result of the propagation of the variation of those answers.
> Prompt optimization: We queried all the models using a zero-shot prompting approach where the factual and counterfactual questions are written in a way that is consistent with each other. We did not perform any optimization of the prompt. However, to guarantee fairness in the experiments we used the same approach for all models and experiments. Therefore, there is no bias towards GPT4 in the experiments. Instead, we observed that this model is better at providing factual and counterfactual answers that its predecessors.
>
> *2. Interpreting decision making:* We thank the reviewer for this comment. We believe that focusing on PN and PS helps to understand to what extent an LLMs is answering a question by composing basic elements of the solution or by simply memorizing answers. Guaranteeing that predictions are achieved by means of a correct reasoning process enhances robustness in the answers because they need to be grounded in a correct real-world reasoning model. Focusing only on predictions may lead to drawing the wrong conclusion that the model ‘understands’ the world when it is merely replicating patterns in the training data, which also leads to hallucinations.

---

> > ### Comment · Reviewer_1YAi · 2024-08-13
> >
> > Thank you for the detailed rebuttal and the additional experiments. Although the framework's reliance on binary variables may limit its broader application, the proposed approach could inspire similar research, given how unexplored evaluating reasoning in LLMs is (besides accuracy on reasoning tasks). Regarding prompt optimization, my concern was that GPT-4 might perform better because it can handle various prompt formats, suggesting that "better" results might be achievable for other (smaller) models with different prompts. I have increased my score to 6.

---

### Author Rebuttal · Authors · 2024-08-06

We thank the four reviews for the positive feedback and the comments that have helped to improve our work. It is encouraging to see that the reviewers find our approach to offer "... a nuanced evaluation that extends beyond the accuracy metric." and that they agree with us that ''...The examination of the concepts of necessity and sufficiency is well-founded, as these are crucial elements in causation and logical reasoning tasks...". We also believe that this work is the right step toward evaluating reasoning in LLMs.

We have added a rebuttal for each reviewer that cover all the points in the discussion. We hope our answers will clarify all the reviewers questions and concerns. To complement some of these answers we have added an extra page of material with another two experiments. The first one shows that our approach can be easily extended to other families of models beyond GPTs. The second experiment demonstrates that counterfactuals are an effective way of testing the models in scenarios not seen an training time. We show that by fine-tuning the models with counterfactual data (most likely unseen at training time but presented to the models in the fine-tunning process) the models performance increases in their approximations to PN and PS.

We are confident that these additions will further substantiate the robustness and utility of our approach, and we look forward to the continued discussion and feedback from the reviewers. Thank you for your valuable feedback and consideration.

---

### Decision · Program_Chairs · 2024-09-25

**Decision:**

Accept (poster)

**Comment:**

Reasoning capabilities of large language models is a topic of great interest to the larger NeurIPS community. This paper provides a novel framework that draws from concepts from causal reasoning to identify whether LLMs are actually able to perform reasoning. The introduction of such a systematic framework to this space is a meaningful contribution to this space, which is otherwise filled with papers that look at shallower metrics like accuracy. However, the primary concern multiple reviewers have raised is the authors’ choice to focus the evaluation on a specific class of reasoning problems. While from the point of view of a conference paper, the choice to focus on a specific class is reasonable, it would be helpful to update both the title and abstract to accurately reflect this limitation. This is also important because the current formulation also limits its focus to problems with boolean variables. Additionally, reviewers have pointed out a lack of a more detailed discussion about the implications of the current results. If accepted, the authors will have access to an additional page. It is recommended that at least part of that space is used to incorporate discussion along these lines.

Given the relevance of the paper and its contribution, I would recommend the paper be accepted.